# Lions *Panthera leo* Prefer Killing Certain Cattle *Bos taurus* Types

**DOI:** 10.3390/ani10040692

**Published:** 2020-04-16

**Authors:** Florian J. Weise, Mathata Tomeletso, Andrew B. Stein, Michael J. Somers, Matt W. Hayward

**Affiliations:** 1Eugène Marais Chair of Wildlife Management, Mammal Research Institute, Department of Zoology and Entomology, University of Pretoria, Pretoria 0002, South Africa; florian.weise@gmail.com (F.J.W.); michael.somers@up.ac.za (M.J.S.); 2CLAWS Conservancy, 32 Pine Tree Drive, Worcester, MA 01609, USA; clawsconservancy@gmail.com (M.T.); andrewstein@clawsconservancy.org (A.B.S.); 3Ongava Research Centre, Private Bag 12041, Ausspannplatz, Windhoek 9000, Namibia; 4Landmark College, 19 River Road South, Putney, VT 05346, USA; 5Centre for Invasion Biology, University of Pretoria, Pretoria 0002, South Africa; 6School of Environmental and Life Sciences, University of Newcastle, University Drive, Callaghan NSW 2308, Australia

**Keywords:** prey preferences, predator–prey interactions, livestock, feeding ecology, domestication, conflict, evolution, domestication, antipredator strategies, unnatural selection

## Abstract

**Simple Summary:**

Livestock production continues to increase throughout the world. Meanwhile, lions are becoming increasingly endangered, in part due to the severe conflict arising from cattle predation. Because cattle closely resemble the body size, shape and herding patterns of wild lion prey, it becomes imperative to understand lion preferences for specific cattle to enhance our ability to design appropriate predation mitigation measures, reduce conflict, and ultimately improve the conservation of lions. Investigating cattle predation patterns in Botswana’s Okavango Delta, we found that lions mostly killed cattle at night and targeted the easiest prey, such as cattle without horns. However, lion preferences differed according to hunting circumstances. Where cattle were confined in enclosures, lions preferred young inexperienced calves, often leading to considerable losses in a single incident. When cattle were left out grazing unprotected, lions preferentially killed cattle with mixed coat patterns and lone males. Losses to lions were driven by cattle characteristics associated with easy husbandry, resulting from domestication. Widespread cattle availability and cavalier protection efforts further fuel losses because cattle no longer possess the key features that enabled their ancestors to coexist with large predators. Cattle are now reliant upon humans to perform critical protection activities.

**Abstract:**

Lion predation on cattle causes severe human–wildlife conflict that results in retaliatory persecution throughout the lion’s geographic range. Cattle closely resemble the body size, shape, and herding patterns of preferred lion prey species. We studied cattle depredation patterns in Botswana’s Okavango Delta and tested whether lions exhibited specific preferences based on cattle demographic characteristics (sex and age), as well as morphological traits (body mass, horn length, and pelage patterns). We also tested whether human disturbance of kills influenced lion energy intake and whether depredation circumstances influenced loss levels. Lions predominantly killed cattle at night (87.1%) and exhibited no preference for either sex. Overall, bulls and calves were most preferred, whereas heifers were significantly avoided, as were cattle with uniform colour patterns. Cattle with mottled pelage patterns were most preferred, especially among free-roaming herds. Preferences were context-specific, with lions preferring inexperienced calves during enclosure attacks (including multiple cases of surplus killing) and free-roaming bulls and oxen. About 13% of adult cattle had no horns, and these were preferentially targeted by lions, while cattle with short horns were killed in accordance with their availability and long horned cattle were highly avoided. The contemporary morphology of Tswana cattle that resulted from unnatural selective pressures during domestication does not offer effective antipredatory protection. Human disturbance of feeding soon after kills occurred reduced cattle carcass consumption by >40% (or about 30 kg per carcass per lion). Lions killed significantly more cattle in nonfortified enclosures than in the veldt, although this was influenced by surplus killing. Our results suggest that cattle predation by lions is driven by availability and cavalier husbandry practices, coupled with morphological features associated with facilitating easy husbandry. Cattle no longer exhibit the key features that enabled their ancestors to coexist with large predators and are now reliant upon humans to perform critical antipredator activities. Hence, the responsibility for mitigating human–wildlife conflict involving lions and cattle lies with people in either breeding traits that minimise predation or adequately protecting their cattle.

## 1. Introduction

Lions *Panthera leo* are specialist predators of prey weighing between 190 and 550 kg [1] and have an accessible prey weight range of 32 to 632 kg [2]. One species they significantly prefer is Cape buffalo *Syncerus caffer* [1,3,4]—a large-bodied bovid occurring in social herds [5]. Given the similarity in morphology and behaviour between buffalo and domestic cattle *Bos taurus* (Figure 1), it is highly likely that lions will also preferentially target the latter. Cattle also exhibit similar seasonal habitat selection patterns [6] and provide easily accessible and stable prey biomass in mixed livestock–wildlife systems [7], exposing them to high predation risk wherever their distribution overlaps with lions. 

Cattle are the primary prey of lions around Gir Protected Area, India [8]. Lion predation on cattle is also a primary source of conflict throughout African range states [9,10,11,12,13], contributing to the lion’s continent-wide decline. Increasing the vulnerability to predation by lions, selective breeding during the domestication of cattle probably led to the loss of traits that their ancestors (aurochs, *Bos primigenius*) used in avoiding predators, such as large body mass, large horns, aggressive nature and perhaps even camouflaged pelage colouration, as traits associated with easy husbandry were selected [14] (Figure 1). As cattle numbers will keep rising throughout Africa [15], it becomes imperative to understand lion preferences for specific cattle to enhance our ability to design appropriate depredation mitigation measures, reduce conflict, and ultimately improve the conservation of lions.

This study aimed at determining whether lions exhibited specific cattle predation preferences based on morphological traits such as sex, horn size, age (as a proxy for body size), and colouration (as a proxy for camouflage). We also tested whether human disturbance of livestock kills influenced the energy intake of lions and whether the circumstances of cattle depredation influenced loss levels.

## 2. Methods

Ethics statement: We conducted research under permit numbers EWT 8/36/4 XXVII (61) and EWT 8/36/4 XXXVIII (63) granted by the Ministry of Environment, Wildlife and Tourism in Botswana. We interviewed human subjects and monitored livestock and lions with ethics approval from the University of Pretoria (no.: EC170525-120, EC170525-120a) and the Institutional Animal Care and Use Committee of the University of Massachusetts (Protocol no.: 2014-0083). The data that support the findings of this study are available from the corresponding author upon reasonable request.

### 2.1. Study Site

Between June 2016 and June 2018, we studied livestock depredation in the eastern Panhandle region of the Okavango Delta (Figure 2), a large inland wetland in Botswana that is characterised by annually variable seasonal flooding [21]. Along the Delta’s periphery, livestock are predominantly free-ranging across unrestricted communal pastures that include a diverse mix of dry savannah woodlands and wetland habitats [22]. This ecosystem mosaic provides critical functional heterogeneity of seasonal habitats for wild and domestic herbivores [6,23]. The Delta also supports one of the last strongholds of free-ranging lions [24,25] and forms part of the Kavango Zambezi Transfrontier Conservation Area. Our study area encompassed five main villages and 44 remote cattle posts located along the boundary of NG/11 and NG/12 communal multiuse areas (Figure 2) that are characterised by settlement, subsistence agriculture, livestock rearing, and nonconsumptive wildlife tourism. From randomised herd counts, we estimated a total standing herd of approximately 16,500 cattle in this area in 2017. Livestock owners received 100% financial compensation for lion-induced stock losses [26], although often with significant delays [27]. Cattle management in the area is highly variable but generally characterised by minimal day-time herding (<10%) and irregular night-time confinement (~40%) in predator-proof enclosures [13]. In the northern Okavango Delta, Tswana beef cattle (a Sanga breed) exhibit strong seasonal habitat selection patterns, preferring woodland habitats (with more digestible grasses) during the wet season and wetland habitats (with more reliable forage and water availability) during the dry season [6]. Lions frequently kill unguarded cattle, resulting in annual compensation claims of approximately 40,400 USD, or about 320 USD per livestock owner, an estimated 4.7% of the annual agropastoral household income in this area [13,27].

### 2.2. Survey Methods

We determined mean cattle herd sizes, horn sizes, and local pelage colouration and coat patterns during randomised herd counts conducted between 2016 and 2018. For cattle older than 18 months, we categorised horn size as “long” (longer than the animal’s ears), “short” (shorter than the animal’s ears), or “absent” (no horns or underdeveloped stumps). We estimated mean body mass for each cattle type and age class from published Tswana cattle weight accounts [18,19,28,29,30] and from 483 local slaughter weight records that we sourced from the main local butchery.

Following voluntary depredation reports by livestock owners and position cluster detection from daily monitoring of GPS-tagged study lions (see [27] for tracking details), we investigated conflict sites and cattle carcasses as soon as possible. Similar to Stander et al. [31], we relied on the expert skills of an indigenous tracker (Christopher Tiro Dimbindo) and two experienced researchers (F.J.W. and M.T.) to determine predator ID, group composition, carcass age, and predation circumstances. We critically assessed the available evidence, including spoor patterns, bite marks, feeding patterns, etc., to verify lion predation. Further, we cross-referenced each conflict site with the GPS location data of telemetered study lions, independently confirming lion predation in 31% of reported cases [27]. We excluded incidents with an unclear case history and carcasses with evident feeding by multiple predator species and those disturbed by carrion feeders such as vultures from our analyses.

For each case, we recorded the date, time and GPS location, and the circumstances of depredation (protective effort, herd size, veldt vs. enclosure location), as well as specific livestock characteristics (age, sex, pelage colouration, and pattern) and financial value. Depending on the prevailing colouration and pattern type, we classified pelage patterns into mutually exclusive categories, being uniform colouration, mixed blotching, and marbled. Cattle that exhibited a blaze but showed no additional markings were classified as uniformly coloured. Three observers (F.J.W., M.T., and expert tracker C.T.D.) classified pelage colouration and pattern independently. In cases of disagreement, we employed a majority rule for final classification. 

Two researchers (F.J.W. and M.T.) independently estimated the carcass-specific consumption of edible body mass (excluding skeleton, hooves and horns, stomach content, and skin) to the nearest 10%. In cases of disagreement, the median value was recorded. Finally, we determined food intake rate (kg) per lion by multiplying the estimated proportion of consumed body mass with the age- and sex-specific Tswana cattle weight estimate and divided the result by the number of lions involved in depredation. Based on how quickly owners retrieved livestock carcasses, observed predator presence at kill sites, and estimated lion spoor age, we distinguished kill sites between “disturbed” (<48 h until carcass removal and/or predators encountered) and “undisturbed” (>48 h until carcass removal and/or no encounter).

### 2.3. Statistics

Due to small sample sizes, we excluded depredation incidents involving goats *Hircus capra* and livestock killed by African wild dogs *Lycaon pictus*, leopards *Panthera pardus*, and spotted hyenas *Crocuta crocuta* from the analyses. We calculated Jacobs’ index [32], D = (*r* − *p*)/(*r* + *p* − 2*rp*), for cattle of differing age class, horn size, sex, and colouration, where *r* represented the proportion of cattle of these categories that were killed and *p* represented the proportion found in the herds they came from. The resulting values range between +1 and −1, where +1 indicates maximum preference and −1 indicates maximum avoidance. Following Hayward et al. [33], we used *t*-tests (for normally distributed data) and binomial sign tests (for non-normally distributed data) to determine if there were significant preferences or avoidance for each category against a mean of 0. Because cattle availability, hunting conditions, and the associated stimuli, may differ considerably between free-range conditions and predation at night enclosures near human habitation, we also calculated context-specific Jacobs’ electivity index values for different cattle types and pelage patterns. Following Clements et al. [2], we also plotted a segmented model to determine the effect of body mass on prey preferences using the *segmented* package in *R* [34]. Segmented, or piecewise, regression is a way to determine abrupt changes in the response variable’s gradient by partitioning the independent variable into discrete groups. We used one-tailed two-sample *t*-tests to compare mean carcass consumption rates between disturbed and undisturbed depredation sites and to compare mean kill frequencies between veldt and enclosure incidents. We report all means ± one standard error (S.E.).

## 3. Results

Between June 2016 and June 2018, mean cattle herd size was 51.0 ± 3.4 animals (range: 2–232, *n* = 181). We investigated a total of 143 cattle depredation incidents by large carnivores, recording details for 197 killed cattle from herds located across five villages and 32 remote cattle posts (Figure 2). Lions caused 163 (82.7%) of these mortalities during 117 incidents, with African wild dogs (12.2%), spotted hyenas (4.1%) and leopards (1.0%) responsible for the remainder. Investigations represented 46% of all carnivore-induced livestock kills reported for government compensation during the same time [27]. 

Lion predation on cattle was mainly concentrated during dark night hours, with 87.1% of incidents (*n* = 93) occurring between dusk and dawn (Figure 3). Enclosure-related depredation (*n* = 30 incidents) occurred exclusively at night, whereas lions sporadically killed free-roaming cattle during day-time hours too. Lions killed a median and modal number of 1.0 cattle per incident (range: 1–7). There were ten incidents, however, in which lions killed more than two cattle (4.20 ± 0.36), all of which occurred in non-predator-proof night enclosures. Of these 42 mortalities, lions left 13 carcasses untouched (31.0%), even when undisturbed, suggesting surplus killing. The mean cattle kill rate during enclosure-related incidents (2.2 ± 0.3 cattle, *n* = 31) was significantly higher than for incidents under free-range conditions (1.1 ± 0.0 cattle, *n* = 86, *t* = 6.12, d.f. = 1, *p* < 0.001). The mean number of lions involved in enclosure-related incidents (2.90 ± 0.37, *n* = 31) was significantly higher than the mean lion group size during free-range depredation incidents (2.25 ± 0.14, *n* = 77, *t* = 2.05, d.f. = 1; *p* = 0.021).

Overall, heifers were significantly avoided (binomial sign test *n* = 20, *p* < 0.001), whereas no other age class was significantly preferred or avoided (Figure 4). Interestingly, however, bulls and calves were most preferred and may become significantly preferred with a larger sample size (Figure 4). Overall, there was no preference for either sex (Figure 4). 

Context-specific preference analysis (Figure 5) confirmed overall preference patterns (Figure 4) but revealed important nuances. Lions demonstrated a strong preference for killing calves during enclosure-related incidents, whereas heifers and cows were strongly avoided in both kraals and beyond. The strong preference for killing bulls in enclosures represents a statistical outlier due to small sample size (*n* = 6). Conversely, lions avoided calves under free-range conditions but exhibited a clear preference for oxen and bulls. Lions did not prefer specific pelage patterns during enclosure-related incidents but showed a clear preference for mixed pelage patterns (blotched and marbled) when killing free-roaming cattle.

Of 725 randomly assessed Tswana cattle >18 months of age (612 females, 113 males), 69.3% had long horns, 17.5% had short horns, and 13.2% exhibited no horns. Male cattle exhibited a significantly higher proportion of long horns (χ^2^ = 16.2, d.f. = 5, *p* < 0.001). Based on 97 adult cattle kills (56 females, 41 males) from herds with verifiable horn lengths, lions preferentially depredated on cattle without horns (*D* = 0.80), killed short horned cattle in accordance with their abundance (*D* = −0.09), and avoided long horned cattle (*D* = −0.79). Long horns result in a 75% reduction in expected kills, while an absence of horns results in a 276% increase in expected kills. 

Cattle with a black and brown mottled pelage were most preferred, while pure black, pure white, and dark brown cattle were all significantly avoided (binomial *n* and *p*, respectively: 17, <0.001; 13, 0.006; 23, <0.001; Figure 4). The segmented model showed no significant changes in slope (Figure 6).

We classified 73 (44.8%) carcasses as disturbed by human activity, 88 (54.0%) as undisturbed, and two (1.2%) as unknown. The mean consumption rate by lions for disturbed carcasses (30.3 ± 3.6%) was significantly lower than that for undisturbed carcasses (71.5 ± 3.4%; *t* = −8.33, d.f. = 1, *p* < 0.001). Calibrated by the average weight per cattle type (see Figure 6) and the number of lions involved in each depredation incident, the body mass consumed per lion from disturbed carcasses (43.23 ± 6.75 kg, *n* = 67) was significantly lower (*t* = 2.91, *p* = 0.002) than that consumed from undisturbed carcasses (73.90 ± 7.75 kg, *n* = 82).

## 4. Discussion

As cattle occupy the centre of the lion’s preferred prey weight range, it is no surprise that their predation is a major source of human–wildlife conflict in Africa. Domestication has led to the loss of a suite of behavioural and morphological characteristics that would have minimised depredation—notably heightened aggression, large body size, and large horns—in favour of traits that assist husbandry practices and easy handling by pastoralists. Today, domestic cattle rely almost entirely upon human pastoralists and their various protective strategies for safeguarding from predators. This may be satisfactory where humans have extirpated large predators; however, the contemporary suite of morphological features that Tswana cattle exhibit in southern Africa does not offer effective protection from lions. These same problems face cattle (and other livestock) elsewhere where they coexist with tigers *Panthera tigris* and lions in India, jaguars *Panthera onca* and pumas *Puma concolor* in South America, and wolves *Canis lupus* and brown bears *Ursus arctos* in the Holarctic.

Depredation investigations provided rare evidence for surplus killing by lions [35] in non-predator-proof night enclosures, showing that these settings can lead to elevated losses. During enclosure attacks, cattle stampede, creating a stimulus-rich highly localized hunting environment for lions. The presence of “a great many easy prey” [35] with limited escape opportunities may induce surplus killing, during which lions preferentially targeted young inexperienced calves that were easily caught and subdued. In these circumstances, i.e., panicked stampedes, lions showed no preference for specific pelage patterns. Lion group size and cattle losses were significantly higher for enclosure-related depredation incidents when compared with free-ranging conditions. Larger prides and male coalitions may specifically target nonfortified enclosures to optimise prey catchability and rapid energy intake in an area where nearly 45% of cattle kills are disturbed by or lost to humans. Spatioecologically, preferential depredation of vulnerable cattle in enclosures manifests in repetitive, goal-oriented movements of specific lions toward easy, confined prey (Figure 7, see also [36]), resulting in habitual livestock raiding by individuals [37]. The timing and nature of enclosure-related depredation corroborates East African studies in that lions accessed vulnerable domestic prey in kraals when human activity was lowest (21:00–06:00), thus minimizing the risk of persecution near human settlements [36,38]. Weise et al. [13] showed the effectiveness of fortified enclosures to prevent lion attacks reliably, but pastoralists did not ubiquitously use these. That most losses occurred while cattle roamed unguarded away from homesteads at night is consistent with other communal grazing areas in southern Africa [39,40]. In Botswana, widespread cattle availability in mixed livestock–wildlife systems during peak lion activity hours [27] enables depredation. While lions may generally exhibit a preference for wild over domestic prey where both are available [41], the stable abundance of cattle in semiarid, seasonally flooded communal pastures in Botswana prompts a behavioural adjustment when migratory wild prey are scarce [7]. Lions then switch their preference to readily available cattle, exacerbating conflict [7]. In the northern Okavango Delta, conflict peaks during low-flood months between September and February when lions have unrestricted access to communal grazing pastures and cattle move farthest into core lion habitat following the availability of critical resources such as grass and water during the late dry season [6,27].

Early disturbance of kill sites by humans reduced carcass consumption by over 40%, the equivalent of a food intake reduction of approximately 30 kg per carcass per lion. This is important because disturbance may thus lead to an increased depredation frequency by lions in communal grazing areas. Lions may kill more frequently to obtain dietary energy balance as they need almost 6 kg of food per day to maintain body condition [42], but populations can average 9.8 kg or more [43,44].

Approximately 31% of adult cattle (mainly females) in the study area lacked or only possessed short horns, limiting their ability to ward off lions. Indeed, lions preferentially predated on animals without horns, thus possibly reducing the risk of injury. Although sufficient horn length remains unknown, Roberts [45] hypothesised that the evolutionary function of long horns in female ruminants includes effective protection against predators. Our data support this. 

Pure black and white colourations may represent extremes of large *Bos* bovid colouration, and although the novelty of these colours may attract lion attention, it does not result in preferential predation. Dark brown is the typical colouration of many ungulates in Africa [46] and is the most avoided by lions (Figure 4). Melin et al. [47] suggest that zebras are as clearly identified by lions as are other similar-sized ungulates, showing their patterned pelage offers no advantage. We found that nonuniform colour patterns were most preferred by lions signifying that these pelages do not offer protection via camouflage (Figure 4). Lions, like domestic cats, have more rods than cones in their retina [48] and have a *tapetum lucidum* [49], giving them a sight advantage at night. This, however, comes with the cost of not seeing colour as well as, for instance, humans [50]. This explains why mixed pelage cattle do not afford any protection, but not why plain coloured cattle are eaten less. Perhaps, as lion respond rapidly to moving prey but seem to have difficulty seeing stationary animals [50], the mixed pelage pattern helps identify movement and so are attacked more frequently. 

Heifers are likely to benefit from safety in numbers [51], whereas bulls spend more time on their own away from herds in typical bovid fashion [52], rendering them more susceptible to predators. In addition, bulls and oxen are often kept separate from female herds and closer to villages, in some instances being tethered together to minimize their movements, as they are utilised for water transport and ploughing of agricultural fields. Despite their longer horns, bulls and oxen may, therefore, provide easy prey opportunities that yield optimal energetic return due to their larger body mass. Calves may lack the experience required to avoid risky habitats or situations. They are also less likely to cause injuries to attacking lions, and so may be the easiest of targets and increasingly preferred by lions, especially during panicked herd stampedes resulting from enclosure attacks when, contrary to predation on free-ranging herds, lions exhibited a clear preference for inexperienced calves. Weise et al. [6] demonstrated that cattle in the study area do not exhibit a pronounced fear in response to lion presence and predation. Instead, cattle utilise different habitat types according to their seasonal resource needs, mimicking the resource utilisation patterns of buffalo [53], thereby increasing predation risk significantly [6].

The morphological and behavioural adaptations that have been selected for during domestication render most cattle unable to avoid and deter predation by lions, stressing the critical importance of effective protection by humans to prevent conflict [54,55]. It appears that cavalier protection attempts, for instance, through insufficient night confinement, may increase conflict by providing lions with easy prey and predictable depredation opportunities.

This study illustrates that the domestication process has altered predator–prey interactions and how this has occurred. Cattle have lost the key features that enabled their ancestors to coexist with large predators, and now are reliant upon humans to perform critical antipredator activities. Overall, lions exhibited only subtle preferences and commonly killed all types of cattle, emphasising that any unguarded, free-ranging livestock is at risk of depredation.

Unless pastoralists employ adequate protection of their herds, methods to mitigate cattle depredation could include the selective breeding of cattle for larger horns and uniform colouration, although we acknowledge that changed colour patterns were a likely outcome of the artificial selection associated with the domestication process [56]. Furthermore, such breeding may conflict with ease of management for husbandry as cattle are likely to become more difficult to handle with the selection for “wilder” traits.

## Figures and Tables

**Figure 1 animals-10-00692-f001:**
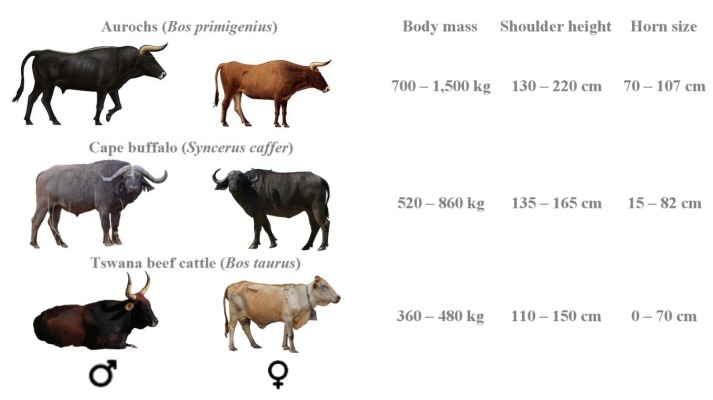
Morphological characteristics of the Aurochs, Cape buffalo, and Tswana beef cattle with adult measurements. Data sources: [14,16,17,18,19], own data. Aurochs graphics retrieved from [20].

**Figure 2 animals-10-00692-f002:**
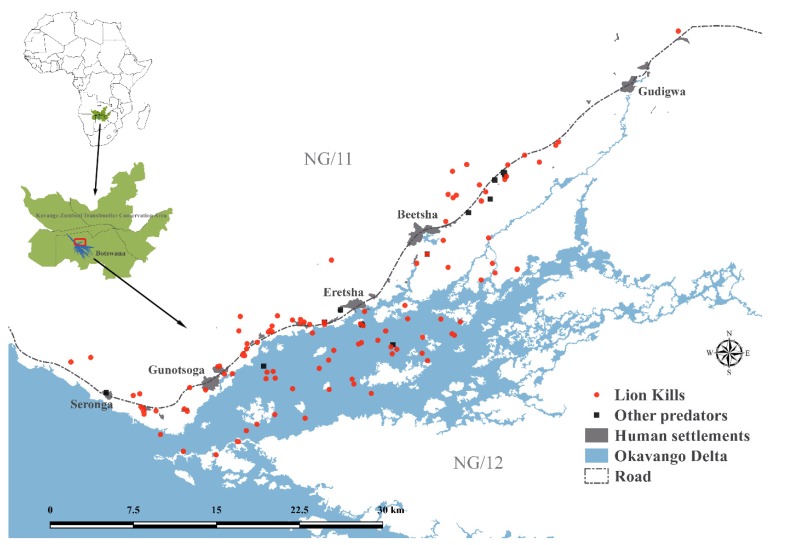
Map showing the distribution of cattle kill sites in northern Botswana. The green inset displays the study area’s location within the Kavango Zambezi Transfrontier Conservation Area.

**Figure 3 animals-10-00692-f003:**
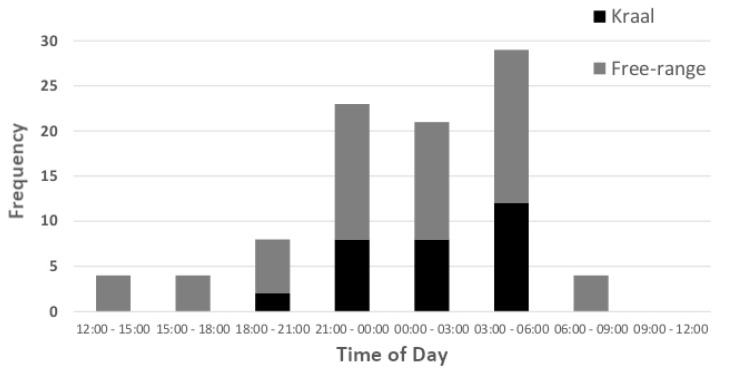
Temporal distribution of 93 cattle predation incidents by lions. Most incidents, 87.1% (*n* = 81), occurred between dusk and dawn. Only cases with known predation times are included.

**Figure 4 animals-10-00692-f004:**
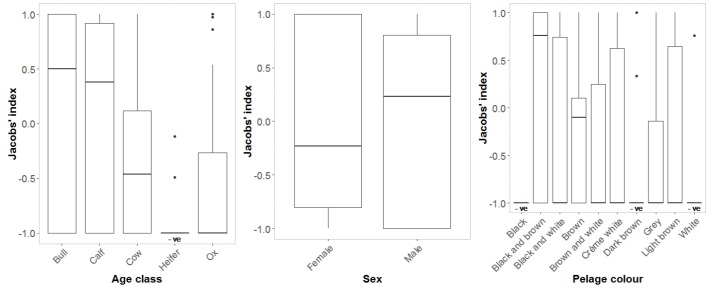
Comparison of mean (± 1 S.E.) Jacobs’ electivity index values for the age class of cattle (type), sex and pelage colouration. Categories that differ significantly from 0 are shown as “+ve” for significantly preferred and “-ve” for significantly avoided. Whiskers show upper and lower quartiles, dots represent outliers.

**Figure 5 animals-10-00692-f005:**
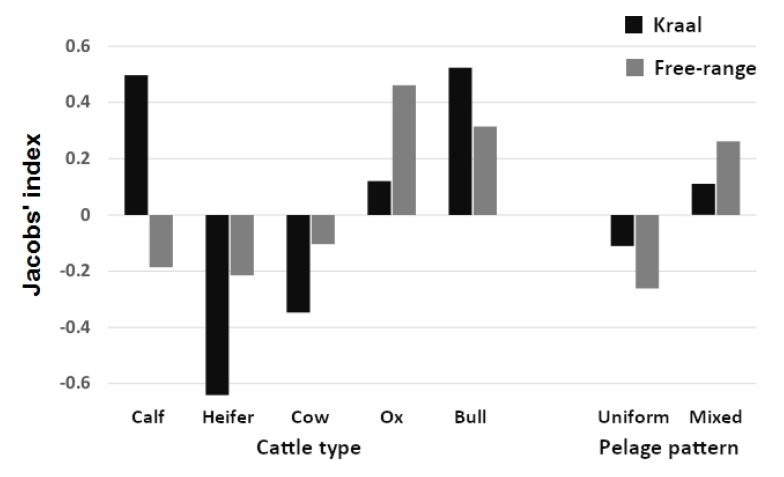
Comparison of context-specific Jacobs’ electivity index values for different cattle types and pelage patterns. Predation context includes free-ranging cattle herds in comparison with cattle confined in night enclosures near villages and cattle posts. “Mixed pelage” includes blotched and marbled coat patterns.

**Figure 6 animals-10-00692-f006:**
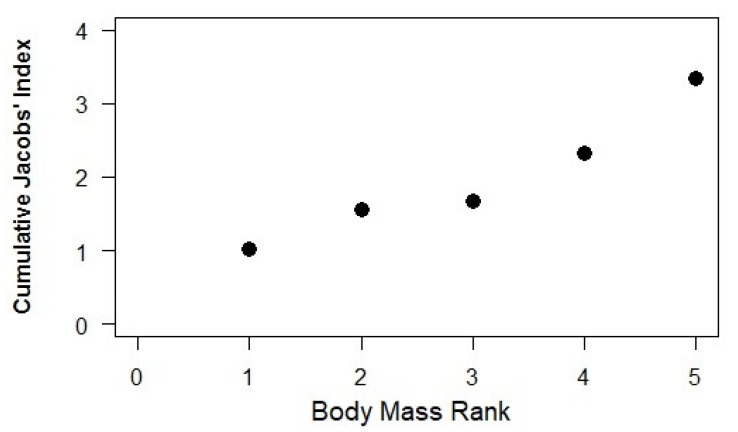
Segmented model plot showing the change in preference for cattle of different weights where 1 represents calves (104 kg), 2–oxen (228 kg), 3–heifers (270 kg), 4–cows (360 kg), and 5–bulls (480 kg).

**Figure 7 animals-10-00692-f007:**
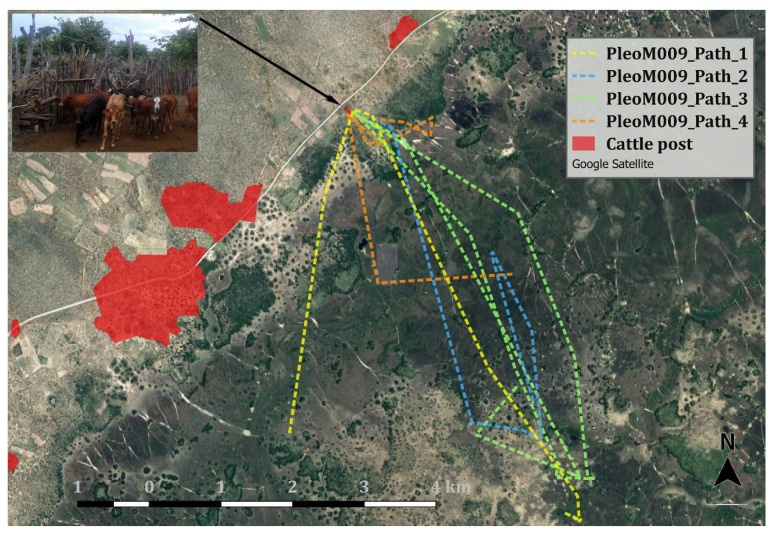
Map showing GPS paths of repeat visitations of male study lion PleoM009 during consecutive cattle depredation incidents at a nonfortified cattle post in 2018. Oriented movements resulted in five attacks during four nights between 18 January and 21 March 2018, with a total loss of seven cattle: four calves, two oxen, and one cow.

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
