# Peer review of "Lions Panthera leo Prefer Killing Certain Cattle Bos taurus Types"

_animals, 2020, doi:10.3390/ani10040692_

Round 1

Reviewer 1 Report

Thank You for the opportunity to review this paper. I read it with interest.  It’s valuable research and important from practical point of view. I have only two suggestions for Authors. I would like to know some details about procedures during carcass inspections. Who is responsible for this job and how they distinguished which cattle was killed by lions or just lions find dead cattle and eat the carrion. I wonder to know if local inhabitants can abuse this procedure to swindle the money. In Discussion some information about eyesight and color differentiation of lion might be useful.

Author Response

Referee 1:

C1 - I would like to know some details about procedures during carcass inspections. Who is responsible for this job and how they distinguished which cattle was killed by lions or just lions find dead cattle and eat the carrion.

Response: We added further detail to the methods. Cattle carcasses were evaluated as soon as possible after livestock owners alerted the researchers about predation incidents, or when the researchers identified a potential kill from GPS location clustering during daily monitoring of lion movements. For this study, we excluded any predation incidents leaving doubt over the nature of events, i.e. had lions indeed killed cattle, or possibly other predators, or did death result from an alternative natural cause? First, during predation site inspections, we ascertained satisfactory evidence of lion predation – for instance, carcasses showed throat or neck bites, which one would not expect to see if cattle had died from disease, for instance, or other causes. Second, in many instances, cattle kills were either observed or over-heard by people, especially kraal-related predation at cattle posts. In these cases, predation sites were investigated immediately the following morning. Third, our indigenous tracking team ascertained that tracks and spoor confirmed a hunting event, i.e. cattle and lion running, a kill site (usually indicated by considerable soil disturbance, blood, and intestines left after lions opened the carcass for feeding), and sometimes also drag marks. Further, we cross-referenced predation incidents with the GPS paths of telemetered study lions and, in most cases, were able to verify which lions had killed cattle. We excluded over 35 cattle predation cases from our analyses for which predation circumstances could not be substantiated beyond doubt. Together, the three investigators, FJW, MT, and Christopher Tiro Dimbindo, have >20 years experience in assessing predation incidents.

C2 - I wonder to know if local inhabitants can abuse this procedure to swindle the money.

Response: Indeed, this is a serious concern for Botswana’s Department of Wildlife and National Parks, which is tasked with investigating claimed predation incidents to establish eligibility for compensation pay-outs. Although this question is beyond the focus of the current paper, we wish to point the reviewer to elaborations on this subject in:

  • WEISE, F. J., HAYWARD, M. W., CASSILLAS AGUIRRE, R., TOMELETSO, M., GADIMANG, P., SOMERS, M. J. & STEIN, A. B. (2018) Size, shape and maintenance matter: a critical appraisal of a global carnivore conflict mitigation strategy - livestock protection kraals in northern Botswana. Conserv., 225, 88-97.

2)      SONGHURST, A. 2017. Measuring human–wildlife conflicts: Comparing insights from different monitoring approaches. Wildl. Soc. Bull., 41, 351-361. https://doi.org/10.1002/wsb.773

Cavalier livestock husbandry in this area, as we report in previous studies, results in a constant un-managed interface of lions with free-roaming cattle. Consequently, a large number of presumed predation incidents are either not assessed rigorously, or even detected. Considering the frequency of cattle predation by lions in this area, the Department of National Parks and Wildlife does not have the resources to investigate each claim in a timely fashion, i.e. before evidence might be disturbed or destroyed. As part of this study, we made case-specific assessment data available to the DWNP in an attempt to improve verification of claimed cattle losses. Songhurst (2017) provides an excellent study of the associated implications for compensation when comparing research-based conflict evaluations with government services. In our study area, it is common that livestock owners will attempt to attribute losses to spotted hyena (no eligibility for compensation) or natural deaths from disease etc (also no eligibility for compensation) to lion depredation in order to fetch maximum compensation payments (only lion-related depredation is compensated at 100% value). Clearly, this is driven by the knowledge that government officers neither have the time nor transport to investigate incidents in a timely fashion to either corroborate or refute claims. This results in a situation in which government compensation funds are regularly exhausted and claims for the last 3 years have not been satisfied due to a shortage of funds. Also, see Weise et al., 2018, 2019a, 2019b

C3 - In Discussion some information about eyesight and color differentiation of lion might be useful.

Response: Excellent point – we combined this comment with Reviewer 2’s comment: Lines 243-247: can the authors speculate why lions avoid uniformly coloured cattle?

Response: We added the following information into the Discussion: “Melin et al. (2016) suggest that zebras are as clearly identified by lions as are other similar-sized ungulates, showing their patterned pelage offers no advantage. We found that non-uniform colour patterns were most preferred by lions signifying that these pelages do not offer protection via camouflage (Fig. 4). Lions, like domestic cats, have more rods than cones in their retina (Ahnelt et al., 2006) and have a Tapetum lucidum (Johnson and Whitteridge, 1968), giving them a sight advantage at night. This, however, comes with the cost of not seeing colour as well as, for instance, humans (Schaller, 1972). This explains why mixed pelage cattle are not afforded any protection, but not why plain coloured cattle are eaten less. Perhaps, as lion respond rapidly to moving prey but seem to have difficulty seeing stationary animals (Schaller, 1972), the mixed pelage pattern helps identify movement and so are attacked more frequently.

These follow the rationale that lions preferred killing cattle with mixed pelage patterns, rather than uniform types, regardless of the actual colouration. Our new Figure 3 shows that nearly all cattle were killed at night. Under low-light circumstances, pelage pattern may be more important for target acquisition than colouration as the pattern creates a discernible contrast in the dark, hence those types (blotched and marbled ones) may simply stand out more under low-light conditions, resulting in higher catchability. Reviewer 2 provided an important corollary remark in that the context of predation (here free-roaming herds vs. kraal confined ones) provide different stimuli. New Figure 5 (context-specific preference evaluation) demonstrates that lions exhibited a weak preference for mixed pelage patterns when attacking kraals, but a much stronger preference for mixed pelage patterns under free-range conditions. It may, therefore, be that contrasting pelage patterns assist lions in singling out a target animal from free-roaming herds under low-light conditions.

Reviewer 2 Report

This is an important contribution to lion conservation, showing that cattle phenotype affects lions’ prey preferences in the Okavango region.  This result may influence cattle breeders’ choice of traits to encourage in their herds (although if all potential cattle prey have long horns and are black, brown, or white, lions may then simply take all morphs equally).  I have only one significant issue, and some minor suggestions.

It would seem that hunting conditions differ considerably between taking cattle in the bush and taking cattle from kraals, presenting the predator with different visual stimuli, different density and availability of potential prey, different patterns of prey escape behaviour, differing proximity to humans, etc.  These all potentially affect lions’ ability to choose an individual to attack and thus their opportunity to show a preference for any of the measured characteristics.   Smaller sample sizes may affect the ability to detect significant differences in preference, but it seems important to break down the data by field or kraal kills.  Further, including the instances of surplus killing adds another confound because preference may not operate in the crush of panicked cattle presenting overwhelming stimuli to the predator.  I encourage the authors to address this issue and present the data by field vs. kraal, even if it is necessary to then lump all data to detect overall preference.

Line 49: What does ‘accessible’ mean in this context?

Line 53: This needs further explanation: does it mean that if both cattle and buffalo are available, lions will prefer cattle?  Why?

Line 143: a brief explanation of segmented models would be helpful

Fig. 3: What do the whiskers and dots represent?

Lines 233-235: Disturbing feeding lions: although chasing lions from a cattle kill probably increases their overall killing rate, a counter argument is that it may also discourage them from taking cattle in future and encourage them to focus on wild prey.

Lines 243-247: can the authors speculate why lions avoid uniformly coloured cattle?

Line 248: might the preference for male cattle be in part because female herds more actively defend against lion attacks, perhaps to protect calves?

The authors may wish to refer to two papers documenting lion movements in relation to bomas in East African rangelands:

Oriol-Cotterill, A., Macdonald , D.W., Valeix, M.,  Ekwanga, S., and  Frank, L.G. 2015. Spatiotemporal patterns of lion space use in a human-dominated landscape. Anim. Behav.  101:27-39. doi:10.1016/j.anbehav.2014.11.020

Suraci, J., Frank, L.G., Oriol-Cotterill, A., Ekwanga, S., Williams, T., Wilmers, C. 2019 Behavior-specific habitat selection by African lions may promote their persistence in a human-dominated landscape. Ecology.   https://doi.org/10.1002/ecy.2644

Author Response

Referee 2:

C1 - It would seem that hunting conditions differ considerably between taking cattle in the bush and taking cattle from kraals, presenting the predator with different visual stimuli, different density and availability of potential prey, different patterns of prey escape behaviour, differing proximity to humans, etc.  These all potentially affect lions’ ability to choose an individual to attack and thus their opportunity to show a preference for any of the measured characteristics.   Smaller sample sizes may affect the ability to detect significant differences in preference, but it seems important to break down the data by field or kraal kills.  Further, including the instances of surplus killing adds another confound because preference may not operate in the crush of panicked cattle presenting overwhelming stimuli to the predator.  I encourage the authors to address this issue and present the data by field vs. kraal, even if it is necessary to then lump all data to detect overall preference.

Response: Excellent comment, Thanks. We conducted the suggested context-specific analysis, which, indeed, revealed important nuances in terms of cattle type and pelage pattern preferences. We added new Figure 5 showing the results. We also added another paragraph summarising these results. “Context-specific preference analysis (Fig. 5) confirmed overall preference patterns but revealed important nuances. Lions demonstrated a strong preference for killing calves during enclosure-related incidents, whereas heifers and cows were strongly avoided in both contexts. The strong preference for killing bulls during enclosure-related represents a statistical outlier due to small sample size (n = 6). Conversely, lions avoided calves under free-range conditions but exhibited a clear preference for oxen and bulls. Lions did not exhibit any preferences for specific pelage patterns during enclosure-related incidents, whilst showing a clear preference for mixed pelage patterns (blotched and marbled) when killing free-roaming cattle.” Additional explanations on these results were added into the Discussion.

C2 - Line 49: What does ‘accessible’ mean in this context?

Response: “Accessible” here refers to the catchable live prey weight range for the species, including the lower and upper bounds of documented lion prey, and thus also including prey range values that are not normally selected for, but have been documented. At the lower and upper bounds, prey is accessible (catchable and consumable) but not generally targeted by lions, either because energetic return on hunting investment is too low (lower bounds, i.e. small prey items such dwarf antelope, juvenile ungulates etc.) or catchability decreases significantly (upper bounds, i.e. prey items such as elephant and other mega-herbivores), hence lions prefer prey between 190-550kg body mass with an optimum energetic return considering hunting risk and investment. Note that these prey weight ranges refer to live prey, not carrion. The term “accessible”, being synonymous for catchable, here seems clear to us and we did not change wording.

C3 - Line 53: This needs further explanation: does it mean that if both cattle and buffalo are available, lions will prefer cattle?  Why?

Response: Good point, the statement is confusing in terms of whether lions may prefer one over the other, which is not what we intended to say. To clarify this, we have rephrased this sentence to read: “... is highly likely that lions will also preferentially target the latter.”

C4 - Line 143: a brief explanation of segmented models would be helpful

Response: additional information on this method added: “Following Clements et al. (2014), we also plotted a segmented model to determine the effect of body mass on prey preferences using the segmented package in R (Muggeo, 2008). Segmented, or piecewise, regression is a way to determine abrupt changes in the response variable’s gradient by partitioning the independent variable into discrete groups.

Segmented regression is useful when the independent variables, clustered into different groups, exhibit different relationships between the variables in these regions. In our case, segmented modelling is a useful way of determining changes in gradient that illustrate body masses the are increasingly or decreasingly preferred by lions.

C5 - Fig. 3: What do the whiskers and dots represent?

Response: Figure is a standard box plot - requested information added to Figure caption: Figure 4. Comparison of mean (± 1 S.E.) Jacobs’ electivity index values for the age class of cattle (type), sex and pelage colouration. Categories that differ significantly from 0 are shown as ‘+ve’ for significantly preferred and ‘-ve’ for significantly avoided. Whiskers show upper and lower quartiles, dots represent outliers.

C6 - Lines 233-235: Disturbing feeding lions: although chasing lions from a cattle kill probably increases their overall killing rate, a counter argument is that it may also discourage them from taking cattle in future and encourage them to focus on wild prey.

Response: A reasonable counter-assumption of course, but in this particular area unlikely to be the case. For instance, several telemetered study lions have continued to prey on cattle for many years, despite having been chased from kills several dozen times. One particular female, for instance, has previously been shot at (and still has shotgun pellets lodged in her shoulder) but continues to raid cattle at considerable rates, rearing successive litters in close proximity to cattle posts and specialising on domestic prey during these times (see Weise et al., 2019b). Furthermore, if chasing lions from kills would indeed result in behavioural adjustment and a preference switch to wild prey (which are abundantly available in the area), we would expect a decrease in cattle depredation incidents. Please refer to Weise et al. (2109a) for supporting data. “While lions were reported to kill only 11 cattle in 2010, compensation claims increased by 2300% to 264 cattle in 2017 (Department of

Wildlife and National Parks, Seronga office).” This trend cannot be simply explained by a simultaneous increase in the local lion and cattle population. In this particular study area, however, it remains unknown whether lions prefer wild prey over cattle in general. This needs further investigation. The rate of cattle killing for several years, notably by a relatively small number of lions, suggests a balanced scenario that is partially driven by easy access and widespread availability of unprotected domestic prey.

C7 - Lines 243-247: can the authors speculate why lions avoid uniformly coloured cattle?

Response: We conflated this with Reviewer 1’s comment 3 – see above response.

C8 - Line 248: might the preference for male cattle be in part because female herds more actively defend against lion attacks, perhaps to protect calves?

Response: In a previous study (Weise et al., 2019a) we found that cattle do not obey any landscape of fear, neither in terms of avoiding lions in space nor in time, and regardless of high depredation rates. Clearly an effect of domestication. Cattle predominantly utilise the landscape according to their seasonal resource needs, not in response to lion presence or predation risk. Additional, though limited, field observations of lions hunting cattle suggest no defense response by adult females – rather, herds exhibit a panicked flight response. However, we cannot ascertain the latter empirically due to a limited number of reliable observations.

Herds mainly appear to benefit from safety in numbers rather than from an active defense against lion attacks. Also, based on the fact that oxen and bulls have longer horns and are heavier than females, we would indeed expect a stronger defense response from males rather than females, most of which have short or no horns. We speculate that, under free-range conditions specifically, bulls and oxen are preferred by lions because:

  1. They are kept alone or in pairs in the lagoons and floodplains near villages because they are regularly used for pulling water and ploughing agricultural fields, thus providing easier hunting targets for lions – instead of selecting and isolating a single target animal from a large free-roaming herd, lions may indeed find it easier to approach and catch single males;
  2. They provide optimum energetic return in terms of consumable biomass; and

C9-The authors may wish to refer to two papers documenting lion movements in relation to bomas in East African rangelands:

Oriol-Cotterill, A., Macdonald , D.W., Valeix, M.,  Ekwanga, S., and  Frank, L.G. 2015. Spatiotemporal patterns of lion space use in a human-dominated landscape. Anim. Behav.  101:27-39. doi:10.1016/j.anbehav.2014.11.020

Suraci, J., Frank, L.G., Oriol-Cotterill, A., Ekwanga, S., Williams, T., Wilmers, C. 2019 Behavior-specific habitat selection by African lions may promote their persistence in a human-dominated landscape. Ecology.   https://doi.org/10.1002/ecy.2644

Response: Very useful materials indeed. We’ve included reference to these studies in the Discussion. It would thus also seem that the finer spatiotemporal patterns that lions exhibit are consistent across bio-geographic regions. Addition to our Discussion: “The timing and nature of enclosure-related depredation corroborates East African studies in that lions accessed vulnerable domestic prey in kraals when human activity was lowest (21:00 – 06:00), thus minimizing the risk of persecution near human settlements (Oriol-Cotterill et al., 2015, Suraci et al., 2019).

Round 2

Reviewer 2 Report

Thank you for attending to my suggestions.  I am happy with the revisions and think this is a very important addition to the lion depredation literature.  

Minor point: I am surprised by the low number of hyaena incidents, especially in poorly protected kraals.  Are hyaenas relatively uncommon in this area or do they just avoid cattle?